# CorDA: Context-Oriented Decomposition Adaptation of Large Language Models for Task-Aware Parameter-Efficient Fine-tuning

Yibo Yang[1], Xiaojie Li[2,3], Zhongzhu Zhou[4], Shuaiwen Leon Song[4], Jianlong Wu[2]

Liqiang Nie[2,†], Bernard Ghanem[1,†]

[1]King Abdullah University of Science and Technology (KAUST)
[2]Harbin Institute of Technology (Shenzhen)   [3]Peng Cheng Laboratory   [4]University of Sydney
† : corresponding authors
https://github.com/iboing/CorDA

## Abstract

Current parameter-efficient fine-tuning (PEFT) methods build adapters widely agnostic of the context of downstream task to learn, or the context of important knowledge to maintain. As a result, there is often a performance gap compared to full-parameter fine-tuning, and meanwhile the fine-tuned model suffers from catastrophic forgetting of the pre-trained world knowledge. In this paper, we propose **CorDA**, a **C**ontext-**or**iented **D**ecomposition **A**daptation method that builds learnable **task-aware adapters** from weight decomposition oriented by the context of downstream task or the world knowledge to maintain. Concretely, we collect a few data samples, and perform singular value decomposition for each linear layer of a pre-trained LLM multiplied by the covariance matrix of the input activation using these samples. The inverse of the covariance matrix is multiplied with the decomposed components to reconstruct the original weights. By doing so, the context of the representative samples is captured through deciding the factorizing orientation. Our method enables two options, the **knowledge-preserved adaptation** and the **instruction-previewed adaptation**. For the former, we use question-answering samples to obtain the covariance matrices, and use the decomposed components with the smallest $r$ singular values to initialize a learnable adapter, with the others frozen such that the world knowledge is better preserved. For the latter, we use the instruction data from the fine-tuning task, such as math or coding, to orientate the decomposition and train the largest $r$ components that most correspond to the task to learn. We conduct extensive experiments on Math, Code, and Instruction Following tasks. Our knowledge-preserved adaptation not only achieves better performance than LoRA on fine-tuning tasks, but also mitigates the forgetting of world knowledge. Our instruction-previewed adaptation is able to further enhance the fine-tuning performance to be comparable with full fine-tuning, surpassing the state-of-the-art PEFT methods such as LoRA, DoRA, and PiSSA.

## 1 Introduction

Large language models (LLMs) have shown remarkable abilities in a wide range of challenging tasks, including questing-answering [11, 51], common sense reasoning [6], and instruction following [85]. While being powerful, LLMs demand exorbitant computation and memory cost when fine-tuning the whole model on downstream tasks due to the huge model capacity. To enable resource-friendly adaptation on downstream tasks, parameter-efficient fine-tuning (PEFT) methods are proposed to largely reduce the number of trainable parameters, by only fine-tuning the newly added adapters [20, 21, 17] or tokens [31, 37, 53], with the original pre-trained weights frozen.

38th Conference on Neural Information Processing Systems (NeurIPS 2024).

Among these PEFT methods, LoRA [21] is increasingly attractive because it is able to keep the model architecture unchanged after fine-tuning so does not induce extra burden in inference. LoRA suggests that the weight change in fine-tuning presents a low rank structure, and employs low-rank matrices with a low hidden dimension to approximate the adaptation [21]. Following studies introduce adaptive low rank choice among different layers [78, 59, 75], decouple the learning of magnitude and direction [42], combine LoRA with pruning or quantization [10, 65, 38, 77], and further reduce the number of trainable parameters [26, 55]. However, existing studies build learnable adapters without considering any data context. As a result, these initialized adapters are task-agnostic, and may not be the optimal choice for the downstream task to learn. Moreover, even if PEFT methods only train a small number of parameters, the fine-tuned model will still suffer from catastrophic forgetting, losing much of the world knowledge contained in the pre-trained LLM [13, 61, 83]. As shown in our visualization (Figures 4 and 5 in Appendix C) for the covariance matrices with input from different datasets, similar outlier patterns can be observed for inputs belonging to the same kind of task, which implies that the responsive parts of the LLM pre-trained weights are different when different tasks are triggered. Therefore, the adapter should be built upon the components most associated with the task of concern, *e.g.,* a new ability to learn or the QA ability to maintain pre-trained world knowledge.

To this end, we propose a **task-aware PEFT** method named as **CorDA**, based on **C**ontext-**or**iented **D**ecomposition **A**daptation. It adopts the same low-rank design as LoRA [21], namely introducing two low rank matrices for each linear layer as a learnable adapter, but associates the context of world knowledge or fine-tuning task with the process of building these adapters. First, we randomly collect a few data samples and assume that they contain representative context of the corresponding task. For example, the question from a question-answering dataset well indicates the ability of preserving the corresponding knowledge, and the query to write a code carries the context of the coding task. We feed these samples into a pre-trained LLM, and obtain the covariance matrix of the input activation of each linear layer, *i.e.,* $C = XX^T \in \mathbb{R}^{d_{in} \times d_{in}}$, where $X$ is the input of this layer. We then perform singular value decomposition (SVD) for the weight $W \in \mathbb{R}^{d_{out} \times d_{in}}$ multiplied by the covariance matrix, *i.e.,* $\text{SVD}(WC) = U\Sigma V^T$, where $U$ and $V$ are singular vectors and $\Sigma$ is the diagonal matrix with the singular values arranged in descending order. In this way, the representative context expressed by these covariance matrices is able to direct the factorizing orientation. Finally, the inverse of these covariance matrices is multiplied with the decomposed components to hold the same inference result with the original model at initialization, *i.e.,* $\hat{W} = U\Sigma V^T C^{-1}$, where $\hat{W}$ is the weight after decomposition and reconstruction.

Our method supports two optional modes for practitioners, **knowledge-preserved adaptation** and **instruction-previewed adaptation**. LLM fine-tuning on downstream tasks is always accompanied by the damage of world knowledge acquired from massive pre-training data [13]. Our knowledge-preserved adaptation enables to learn new tasks effectively while keeping world knowledge as sound as possible. In this mode, we use questions from question-answering dataset, such as TriviaQA [23] and Natural Questions [27, 29], to obtain the covariance matrices whose pattern corresponds to the LLM ability in retrieving knowledge. After SVD with these covariance matrices, we use the components with the smallest $r$ singular values, *i.e.,* $U_{[:,-r:]}$, $\Sigma_{[-r:]}$, and $(V^T C^{-1})_{[-r:,:]}$ [1], to initialize a learnable low-rank adapter, and the other components that are key to preserving knowledge are frozen. Alternatively, when one only aims to achieve performance as high as possible on the fine-tuning task without concern for world knowledge maintenance, our instruction-previewed adaptation is suggested. In this mode, we use the instruction and response from the fine-tuning task, *e.g.* query to write a code and its answer, to produce the covariance matrices. Similarly, the pre-trained weights will be decomposed in an orientation such that the context of the fine-tuning task dominates in the principal singular values and vectors. Therefore, we use the largest $r$ components, *i.e.,* $U_{[:,:r]}$, $\Sigma_{[:r]}$, and $(V^T C^{-1})_{[:r,:]}$, to initialize a learnable low-rank adapter, with the other components frozen. The adapter built upon the context of the fine-tuning task well accommodates the new ability, and thus leads to a better fine-tuning performance.

Our method brings flexibility in choosing between stronger fine-tuning performance or more preserved world knowledge, and can be adopted according to the practical demand. Both the two modes are as efficient as LoRA [21]. After fine-tuning, the adapter can be merged with the frozen part in the

---

[1] $U_{[:,-r:]}$, $\Sigma_{[-r:]}$, and $(V^T C^{-1})_{[-r:,:]}$ represent the last $r$ columns of $U$, the last $r$ diagonal elements of $\Sigma$, and the last $r$ rows of $V^T C^{-1}$, respectively. $U_{[:,:r]}$, $\Sigma_{[:r]}$, and $(V^T C^{-1})_{[:r,:]}$ represent the first $r$ columns of $U$, the first $r$ diagonal elements of $\Sigma$, and the first $r$ rows of $V^T C^{-1}$, respectively.

same manner as LoRA. In experiments, CorDA in knowledge-preserved adaptation not only enjoys better fine-tuning performance than LoRA on Math [9, 71], Code [8, 3], and Instruction Following [80], but also largely mitigates the deterioration of performance on world knowledge benchmarks including TriviaQA [23], NQ open [29], and WebQS [4]. The instruction-previewed adaptation is able to further strengthen the performance on fine-tuning tasks, surpassing the state-of-the-art PEFT methods including LoRA [21], DoRA [42], and PiSSA [47].

## 2 Related Work

**Parameter-Efficient Fine-Tuning.** Since large language models (LLMs) have tens and even hundreds of billions of parameters [1, 5], full-parameter fine-tuning will cause unbearable computation and memory cost [79, 67]. Parameter-efficient fine-tuning (PEFT) is developed to reduce resource consumption by only fine-tuning a small number of learnable parameters [12, 64]. Adapter-based methods introduce additional modules into LLMs and only fine-tune them during fine-tuning [20, 17, 30, 45, 49, 24]. Another line of research appends extra soft prompts into the input or hidden layers and only train these learnable vectors [31, 37, 53, 84]. However, most of these methods change the model architecture or increase the inference burden. Based on the insight that the weight change after fine-tuning possesses a low rank structure [32, 2], low-rank adaptation (LoRA) [21] proposes to use two low-rank small matrices as the learnable adapter, without modifying model architecture or bringing inference cost after fine-tuning. LoRA has inspired a range of variants that employ adaptive low rank in different layers [78, 59, 75], explore the adapter design [42, 7, 50, 79], combine LoRA with pruning [77], quantization [10, 65, 38], and mixture-of-expert [41, 13], and introduce alternative way to initialize the adapter [47]. Nevertheless, existing PEFT methods rarely consider data context when building the learnable adapter. Data context has been proved to be instrumental in guiding quantization and compression [40, 73, 28]. In our study, we utilize data context for PEFT, building adapters based on context-oriented decomposition to better maintain world knowledge or accommodate new ability.

**Knowledge Forgetting.** Deep learning models are prone to drastically forgetting the acquired knowledge when adapting to a new task, known as catastrophic forgetting [14, 54, 25, 52, 44, 70, 36]. A series of methods has been proposed to mitigate this issue using knowledge distillation [39, 19, 33], rehearsal [56, 69], and dynamic architecture [66]. In the era of large models [62], however, world knowledge is acquired by pre-training on massive data, which could be intractable to re-use in fine-tuning [34, 35]. The huge model capacity also hinders the feasibility of knowledge distillation and dynamic architecture, especially in the case of continuous fine-tuning [16, 74, 57, 15, 22]. Some studies introduce extra LLaMA layers [61] or mixture of experts [13] with the pre-trained layers frozen to strike a balance between keeping world knowledge and learning new tasks. Alternative approaches adopt merging schemes to enable diverse abilities [76, 72, 83]. Different from these studies, our method enables to achieve world knowledge maintaining in the process of parameter-efficient fine-tuning, without changing model architecture or relying on a post-merging step.

## 3 Method

We review the LoRA method in Sec. 3.1. We develop our context-oriented decomposition in Sec. 3.2, which provides the basis of our knowledge-preserved adaptation and instruction-previewed adaptation introduced in Sec. 3.3 and Sec. 3.4, respectively.

### 3.1 Preliminaries on Low-Rank Adaptation

LoRA suggests that the weight change in LLM fine-tuning presents a low rank structure, and thus proposes to use the product of two low rank matrices to learn the weight change with the pre-trained weights frozen during fine-tuning [21]. Given the pre-trained weight $W \in \mathbb{R}^{d_{out} \times d_{in}}$ from an LLM, the weight after fine-tuning can be formulated as:

$$W^* = W + \Delta W = W + BA, \tag{1}$$

where $W^*$ is the weight after fine-tuning, $\Delta W$ is the weight change, and $BA$ is the low rank decomposition of $\Delta W$ into two smaller matrices $B \in \mathbb{R}^{d_{out} \times r}$ and $A \in \mathbb{R}^{r \times d_{in}}$ with an intrinsic rank of $r \ll \min(d_{out}, d_{in})$. In this way, the number of learnable parameters can be largely reduced

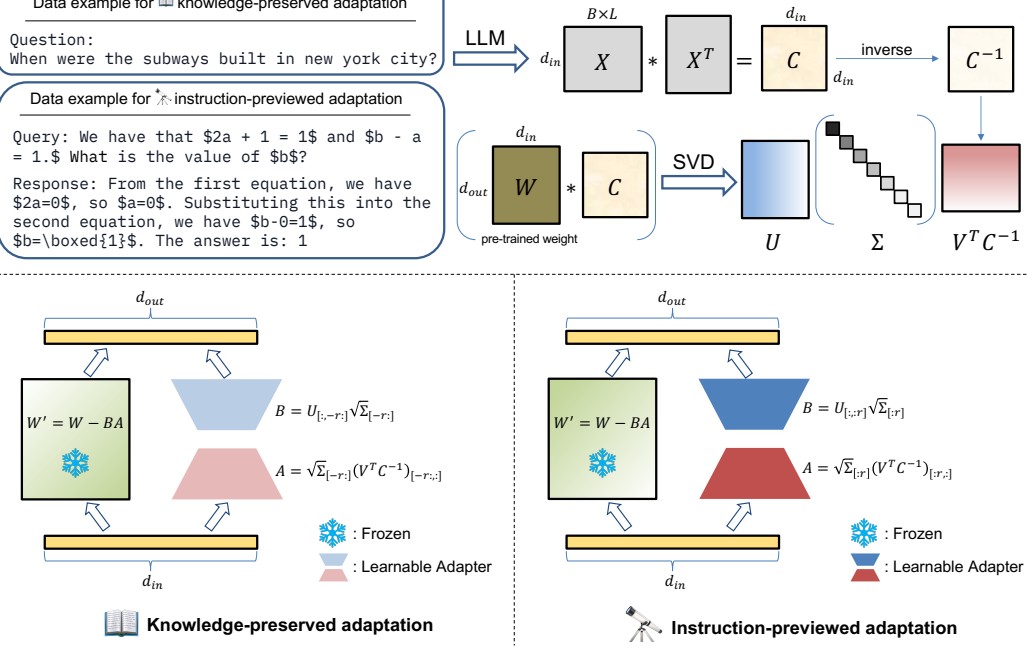

Figure 1: An overall illustration of our proposed method. We perform singular value decomposition oriented by the covariance matrix to aggregate task context into the principal components (up), which are frozen for maintaining world knowledge (down left) or utilized to initialize the learnable adapter for better fine-tuning performance (down right). The dark-colored adapter refers to the components with the largest $r$ singular values, while the light one is composed of the smallest $r$ components.

by freezing the pre-trained weight $W$ and only fine-tuning the matrices $B$ and $A$. LoRA adopts the Kaiming initialization [18] to randomly initialize $A$, and $B$ is initialized as an all-0 matrix such that $\Delta W = 0$ at the start of training to circumvent a deviation from the pre-trained model. After fine-tuning, the learnable adapter $BA$ can be merged into the pre-trained weight $W$ without changing the original model architecture and introducing extra inference burden.

Despite the success of LoRA-based methods, when building the learnable adapter, existing studies widely ignore the data context from the target ability that users are particularly concerned with.

## 3.2 Context-Oriented Decomposition

Pre-trained large language models are endowed with multi-faced abilities, such as answering the question regarding world knowledge, common sense reasoning, and instruction following. When different kinds of input messages are fed into an LLM, *e.g.*, a question in some domain and a query to solve a math problem, even though they are processed by the same pre-trained weights, different abilities are triggered. The covariance matrix of each layer's activation will exhibit different outlier patterns as they are responsive to the task triggered to highlight different aspects of the pre-trained weight. Therefore, the covariance matrix is able to capture task context. Inspired by this insight, we leverage the covariance matrix inside LLM to build adapters catering to a certain ability.

The process of our context-oriented decomposition is shown in Figure 1. First, we randomly collect some samples from the training data of some task with interest, *e.g.*, question answering or Math, and feed these samples into the LLM used to fine-tune. Denote $X \in \mathbb{R}^{d_{in} \times BL}$ as the input activation of a linear layer where $d_{in}$ is the input dimension, $B$ is the number of samples we collect, and $L$ represents the sequence length. We have the covariance matrix $C = XX^T \in \mathbb{R}^{d_{in} \times d_{in}}$. We then perform singular value decomposition for the weight multiplied by the covariance matrix as:

$$\text{SVD}(WC) = U\Sigma V^T = \sum_{i=1}^{R} \sigma_i \mathbf{u}_i \mathbf{v}_i^T, \tag{2}$$

where $W \in \mathbb{R}^{d_{out} \times d_{in}}$ is the weight of this linear layer, $U \in \mathbb{R}^{d_{out} \times d_{out}}$ and $V \in \mathbb{R}^{d_{in} \times d_{in}}$ are orthogonal matrices containing singular vectors $\mathbf{u}_i \in \mathbb{R}^{d_{out}}$ and $\mathbf{v}_i \in \mathbb{R}^{d_{in}}$, $\Sigma \in \mathbb{R}^{d_{out} \times d_{in}}$ is a diagonal matrix with singular values $\sigma_i$ on its diagonal arranged in descending order, and $R$ is the rank (the number of non-zero singular values) of $WC$, *i.e.*, $R \leq \min\{d_{out}, d_{in}\}$.

To not change the inference result at the initialization of fine-tuning, we reconstruct $W$ by:

$$\hat{W} = \texttt{SVD}(WC)C^{-1} = U\Sigma(V^T C^{-1}) = \sum_{i=1}^{R} \sigma_i \mathbf{u}_i \hat{\mathbf{v}}_i^T, \tag{3}$$

where $C^{-1}$ denotes the inverse of $C$, and $\hat{\mathbf{v}}_i^T$ is the $i$-th row vector of $V^T C^{-1}$. In case the covariance matrix $C$ is not invertible, we adopt a strategy to dynamically add positive values on the diagonal elements of $C$ to ensure invertible. Concretely, we multiply a positive coefficient with the average value of the diagonal elements of $C$, and add it on the diagonal. Then we calculate the $\ell_2$ distance between $CC^{-1}$ and an identity matrix. If it is higher than a threshold, we double the coefficient and perform this step again, until the distance reaches below the threshold.

After our context-oriented decomposition, the first several components of $\mathbf{u}_i$ and $\hat{\mathbf{v}}_i$ with the largest singular values $\sigma_i$ depict the dominant characteristics of the task associated with $C$. We can decide either to maintain these key components to not sacrifice the corresponding ability, or to adapt them for better performance on the task, which leads to our two implementation modes in the following two subsections, respectively.

## 3.3 Mode 1: Knowledge-Preserved Adaptation

We introduce knowledge-preserved adaptation that enables to learn new tasks while maintaining world knowledge. In this mode, we use the question data from question-answering training data, such as TriviaQA [23] and Natural Questions [27, 29], to obtain the covariance matrices whose pattern corresponds to the knowledge retrieving ability of the LLM. When fine-tuning on a new task, as shown in Figure 1, we use the last $r$ components with the smallest $r$ singular values in Eq. (3) to build learnable adapters as:

$$W' = W - BA, \quad B = U_{[:,-r:]}\sqrt{\Sigma}_{[-r:]}, \quad A = \sqrt{\Sigma}_{[-r:]}(V^T C^{-1})_{[-r:,:]}, \tag{4}$$

where $B \in \mathbb{R}^{d_{out} \times r}$ and $A \in \mathbb{R}^{r \times d_{in}}$ are the initialized matrices in the learnable adapter, $BA = \sum_{i=R-r+1}^{R} \sigma_i \mathbf{u}_i \hat{\mathbf{v}}_i^T$ corresponds to the last $r$ components in Eq. (3), $\sqrt{\Sigma}_{[-r:]}$ is a diagonal matrix with the squared root of the smallest $r$ singular values on the diagonal, and $W'$ corresponding to the first $R - r$ components in Eq. (3) is frozen during fine-tuning. We have $W'$ as the difference between $W$ and $BA$ instead of summing the first $R - r$ components to avoid the numerical error between $\hat{W}$ and $W$ introduced by the decomposition and inversion operations. After fine-tuning, the learned matrices $B^*$ and $A^*$ can be merged into $W'$ as $W^* = W' + B^*A^*$. This mode is featured by preserving world knowledge as the knowledge retrieving ability captured by the principal $R - r$ components is frozen. It is also more effective than a zero-initialized adapter in learning new abilities as verified by our experiments.

## 3.4 Mode 2: Instruction-Previewed Adaptation

In the circumstance that pursuing a higher performance on the fine-tuning task is the priority, our instruction-previewed adaptation will be favorable. In this mode, we collect instruction and response from the training data used for fine-tuning, *e.g.* the query to solve a math problem and its answer shown in Figure 1 as an example. The prompts are fed into the LLM to produce the covariance matrices whose pattern is associated with the task to learn. We use the first $r$ components with the largest $r$ singular values in Eq. (3) to build learnable adapters as:

$$W' = W - BA, \quad B = U_{[:,:r]}\sqrt{\Sigma}_{[:r]}, \quad A = \sqrt{\Sigma}_{[:r]}(V^T C^{-1})_{[:r,:]}, \tag{5}$$

where $B$ and $A$ are the initialized matrices in the learnable adapter, $BA = \sum_{i=1}^{r} \sigma_i \mathbf{u}_i \hat{\mathbf{v}}_i^T$ corresponds to the first $r$ components in Eq. (3), $\sqrt{\Sigma}_{[:r]}$ is the squared root of the largest $r$ singular values in a diagonal matrix. Similar to knowledge-preserved adaptation, $W'$ containing the remaining $R - r$ components is frozen during fine-tuning. This mode enables the initialized adapters to pre-capture

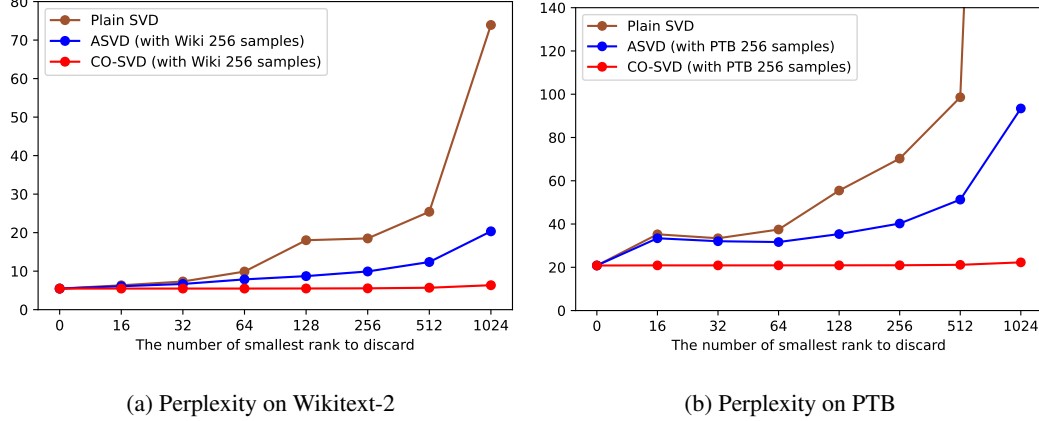

(a) Perplexity on Wikitext-2         (b) Perplexity on PTB

Figure 2: Perplexity (lower is better) on (a) Wikitext-2 and (b) Penn TreeBank (PTB) after decomposing the LLaMA-2-7B weights and reconstruction discarding the smallest $r$ singular values and their singular vectors. We compare our context-oriented decomposition (CO-SVD) with plain SVD and ASVD. The perplexity of plain SVD on PTB at $r = 1024$ is 763.4, which is out of the shown range.

the main characteristics of the fine-tuning task, leading to stronger performance after training. A recently proposed method [47] also performs SVD and uses the first $r$ components to initialize an adapter for fine-tuning. However, their decomposition is agnostic with respect to any data context. Our adapter capturing the task context in advance can well accommodate the new ability and lead to better fine-tuning performances in our experiments.

# 4 Experiments

In experiments, we fine-tune the pre-trained large language model LLaMA-2-7b [58] on Math, Code, and Instruction Following tasks, and also apply our method to the General Language Understanding Evaluation (GLUE) benchmark with RoBERTa$_{base}$ [43]. The world knowledge is evaluated by the exact match scores (%) on TriviaQA [23], NQ open [29], and WebQS [4]. Following the settings in [47], the Math ability is trained on MetaMathQA [71] and tested on GSM8k [9] and Math [71] validation sets. Code is trained on CodeFeedback [82] and tested on HumanEval [8] and MBPP [3]. Instruction following is trained on WizardLM-Evol-Instruct [63] and tested on MTBench [81]. The complete implementation details are described in Appendix A.

## 4.1 Analysis of the Ability to Capture Context

We conduct an experiment to demonstrate the ability of our proposed decomposition method in Sec. 3.2 to capture context in its principal components. We use different methods including the Plain SVD, ASVD [73], and our Context-Oriented SVD (CO-SVD), to perform full decomposition of the LLaMA-2-7B pre-trained weights in all layers, and then discard the smallest $r$ singular values and their corresponding left and right singular vectors to reconstruct the weights for testing.

As shown in Figure 2, as the number of discarded ranks increases, the performances with Plain SVD on both Wikitext-2 [48] and PTB [46] are getting worse steeply. ASVD considers data context using activation absolute mean values, and helps to relieve the deterioration compared to Plain SVD. However, when discarding more than 256 ranks, the Perplexity also diverges sharply. In contrast, our method is able to maintain a stable performance very close to the original pre-trained weights even when the smallest 1024 components are discarded. The result indicates that our method is proficient with aggregating context from limited samples (256 Wiki or PTB samples in this example) into the principal components. It also evidences the potential of our method to maintain important knowledge when freezing the principal components and to learn new abilities when adapting these components. The detailed numbers of the experiment in Figure 2 are listed in Table 6 (Appendix B), where we also test the effect of sample number and dataset choice when collecting the covariance matrices.

Table 1: The experimental results of CorDA in the knowledge-preserved adaptation mode and comparison with full fine-tuning, LoRA, and PiSSA. LLaMA-2-7B is used to fine-tune on (a) Math, (b) Code, and (c) Instruction Following tasks. The rank $r$ of LoRA, PiSSA, and CorDA is 128. CorDA is initialized with the NQ open samples to collect the covariance matrices. All methods are implemented by us under the same training and evaluation settings. The row of "LLaMA-2-7B" shows the world knowledge performance of the original pre-trained model.

(a) Math

| Method | #Params | Trivia QA | NQ open | WebQS | GSM8k | Math | Avg |
|---|---|---|---|---|---|---|---|
| LLaMA-2-7B | - | 52.51 | 14.99 | 5.86 | - | - | - |
| Full fine-tuning | 6738M | 43.64 | 3.13 | 6.35 | 48.90 | 7.48 | 21.90 |
| LoRA [21] | 320M | 44.17 | 1.91 | 6.64 | 42.68 | 5.92 | 20.26 |
| PiSSA [47] | 320M | 39.71 | 1.02 | 6.30 | **51.48** | **7.60** | 21.22 |
| CorDA (ours) | 320M | **44.30** | **9.36** | **7.14** | 44.58 | 6.92 | **22.46** |

(b) Code

| Method | #Params | Trivia QA | NQ open | WebQS | HumanEval | MBPP | Avg |
|---|---|---|---|---|---|---|---|
| LLaMA-2-7B | - | 52.51 | 14.99 | 5.86 | - | - | - |
| Full fine-tuning | 6738M | 29.29 | 8.53 | 3.44 | **25.42** | **25.64** | 18.46 |
| LoRA [21] | 320M | **51.42** | 9.30 | 8.46 | 16.8 | 21.51 | 21.50 |
| PiSSA [47] | 320M | 47.07 | 9.16 | 8.14 | 19.48 | 23.84 | 21.54 |
| CorDA (ours) | 320M | 50.02 | **11.72** | **8.56** | 18.36 | 20.91 | **21.91** |

(c) Instruction Following

| Method | #Params | Trivia QA | NQ open | WebQS | MTBench | Avg |
|---|---|---|---|---|---|---|
| LLaMA-2-7B | - | 52.51 | 14.99 | 5.86 | - | - |
| Full fine-tuning | 6738M | 26.6 | 8.45 | 6.84 | 4.85 | 11.69 |
| LoRA [21] | 320M | 47.46 | 10.28 | 7.73 | 4.60 | 17.52 |
| PiSSA [47] | 320M | 36.76 | 9.67 | 5.86 | 4.92 | 14.30 |
| CorDA (ours) | 320M | **50.34** | **14.43** | **8.17** | **5.05** | **19.50** |

## 4.2 Knowledge-Preserved Adaptation Results

We fine-tune LLaMA-2-7B with full fine-tuning, LoRA, PiSSA, and our proposed CorDA on Math, Code, and Instruction Following tasks. In the knowledge-preserved adaptation mode, we randomly sample 256 questions from the NQ open training set and collect covariance matrices to initialize the adapters by Eq. (4). We report the fine-tuning performance and also the world knowledge performance of the fine-tuned model to manifest the overall ability of both new task learning and world knowledge maintaining. As shown in Table 1a, after fine-tuning on Math, all the three compared methods, full fine-tuning, LoRA, and PiSSA, suffer from drastic performance drop on TriviaQA and NQ open. Especially on NQ open, the ability is almost lost. PiSSA achieves the best accuracies on both GSM8k and Math, but meanwhile has the lowest results on the world knowledge benchmarks among the four methods. As a comparison, our method not only enjoys better fine-tuning performances than LoRA, but also achieves the best results on the three world knowledge benchmarks.

A similar pattern can be also observed in the fine-tuning of Code. As shown in Table 1b, full fine-tuning has the best ability on HumanEval and MBPP, but is the lowest on world knowledge performance. It is understandable that CorDA in knowledge-preserved adaptation is not as advantageous as PiSSA for fine-tuning performance, because PiSSA uses the largest singular values and their singular vectors as the adapter that dominates the weight update, while we keep them frozen and adapt the smallest components. Nevertheless, our method has the best average score in all the three fine-tuning tasks. A surprising result is achieved by our method in instruction following in Table

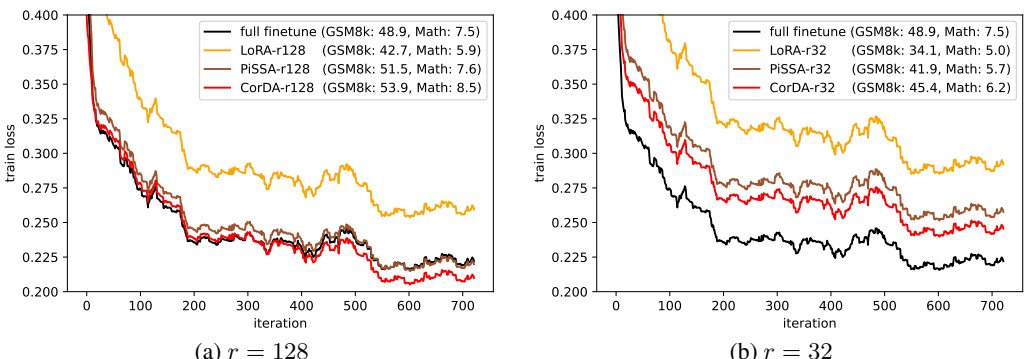

| | (a) $r = 128$ | | (b) $r = 32$ |
|---|---|---|---|

Figure 3: The training loss curves on MetaMath of full fine-tuning, LoRA, PiSSA, and CorDA with (a) rank 128 and (b) rank 32. The corresponding accuracies on GSM8k and Math are reported on the legends. Smoothing is performed for the loss curves.

Table 2: The experimental results of CorDA in the instruction-previewed adaptation mode on Math, Code, and Instruction Following tasks using LLaMA-2-7B. CorDA is initialized with samples from each of the fine-tuning datasets (MetaMathQA, CodeFeedback, and WizardLM-Evol-Instruct) for the three tasks, respectively. The rank $r$ of LoRA, DoRA, PiSSA, and CorDA is 128. All methods are implemented by us under the same training and evaluation settings.

| Method | #Params | GSM8k | Math | HumanEval | MBPP | MTBench | Avg |
|---|---|---|---|---|---|---|---|
| Full fine-tuning | 6738M | 48.9 | 7.48 | **25.42** | **25.64** | 4.85 | 22.46 |
| LoRA [21] | 320M | 42.68 | 5.92 | 16.80 | 21.51 | 4.60 | 18.30 |
| DoRA [42] | 321M | 41.77 | 6.20 | 16.86 | 21.60 | 4.48 | 18.18 |
| PiSSA [47] | 320M | 51.48 | 7.60 | 19.48 | 23.84 | 4.92 | 21.46 |
| CorDA (ours) | 320M | **53.90** | **8.52** | 21.03 | 24.15 | **5.15** | **22.55** |

1c, where CorDA surpasses all the three compared methods in both world knowledge performance and the new ability evaluated by MTBench. The world knowledge on NQ open is almost intact (14.43%) compared to the original performance (14.99%) before fine-tuning. These results reveal that our knowledge-preserved adaptation is an effective way to mitigate world knowledge forgetting and improve the overall ability.

## 4.3 Instruction-Previewed Adaptation Results

When the goal is to learn the target task as much as possible without concerning the loss of world knowledge, CorDA in the instruction-previewed adaptation mode satisfies this demand as it is able to further strengthen the fine-tuning performance. In this mode, we randomly sample instruction and response from the training data to collect the covariance matrices, and adopt the largest $r$ components to initialize the adapters by Eq. (5). We compare our method with full fine-tuning, LoRA, DoRA, and PiSSA in the same three tasks, Math, Code, and Instruction Following. Compared with the knowledge-preserved adaptation in Table 1, CorDA in the instruction-previewed adaptation mode largely improves the fine-tuning performance on the five benchmarks, as shown in Table 2. Concretely, our method achieves the best performance on GSM8k, Math, MTBench, and the average score. Compared with PiSSA that adopts a similar adapter design but with no data context, our method has better results on all the five benchmarks. The training loss curves on Math are shown in Figure 3, where CorDA converges at a lower loss than LoRA and PiSSA in both $r = 128$ and $r = 32$. Only CorDA exhibits an obvious lower loss than full fine-tuning when $r = 128$. These results corroborate the benefits of data context to the initialized adapter, *i.e.,* the pre-captured task characteristic is able to accommodate the new ability and lead to a better performance.

We also apply our method to the General Language Understanding Evaluation (GLUE) benchmark [60] by fine-tuning the RoBERTa$_{base}$ model [43]. We adopt LoRA, DoRA, and our method with

Table 3: The experimental results of CorDA in the instruction-previewed adaptation mode on the GLUE benchmark using RoBERTa$_{base}$. CorDA is initialized with samples from each of the fine-tuning datasets. The rank $r$ of LoRA, DoRA, and CorDA is 128. All methods are implemented by us under the same training and evaluation settings. Matthew's correlation and Pearson's correlation are the metrics of CoLA and STS-B, respectively. The metric of the other tasks is accuracy.

| Method | #Params | SST-2 | MRPC | CoLA | QNLI | RTE | STS-B | Avg |
|---|---|---|---|---|---|---|---|---|
| Full fine-tuning | 125M | 93.81 | 88.48 | 59.56 | 92.07 | 74.01 | **90.49** | 83.07 |
| LoRA [21] | 21M | **94.15** | 82.84 | 54.24 | 92.48 | 64.26 | 88.58 | 79.43 |
| DoRA [42] | 21M | 93.58 | 83.58 | 51.93 | **92.59** | 64.98 | 88.71 | 79.23 |
| CorDA (ours) | 21M | 93.12 | **89.71** | **59.60** | 91.49 | **76.17** | 90.17 | **83.38** |

Table 4: Ablation experiments of the data choice used to collect covariance matrices and the adapter building manner in the knowledge-preserved adaptation mode. †: corresponds to the result of PiSSA that performs plain SVD and uses the largest $r$ components to initialize the adapter.

| Method | Context | Adapter | Trivia QA | NQ open | WebQS | GSM8k | Math | Avg |
|---|---|---|---|---|---|---|---|---|
| Plain SVD† | none | largest $r$ | $39.71_{\pm 0.26}$ | $1.02_{\pm 0.23}$ | $6.30_{\pm 0.39}$ | $51.48_{\pm 0.34}$ | $7.60_{\pm 0.18}$ | 21.22 |
| Plain SVD | none | smallest $r$ | $39.94_{\pm 0.17}$ | $4.21_{\pm 0.41}$ | $6.25_{\pm 0.17}$ | $43.29_{\pm 0.37}$ | $5.96_{\pm 0.13}$ | 19.93 |
| CO-SVD | Wikitext-2 | smallest $r$ | $42.93_{\pm 0.13}$ | $7.20_{\pm 0.15}$ | $6.40_{\pm 0.27}$ | $42.99_{\pm 0.34}$ | $5.80_{\pm 0.09}$ | 21.06 |
| CO-SVD | Trivia QA | smallest $r$ | $44.59_{\pm 0.34}$ | $8.86_{\pm 0.20}$ | $7.53_{\pm 0.14}$ | $44.81_{\pm 0.28}$ | $6.84_{\pm 0.16}$ | 22.53 |
| CO-SVD | NQ open | smallest $r$ | $44.30_{\pm 0.22}$ | $9.36_{\pm 0.16}$ | $7.14_{\pm 0.26}$ | $44.58_{\pm 0.33}$ | $6.92_{\pm 0.13}$ | 22.46 |

Table 5: The instruction following performance of CorDA using WizardLM-Evol-Instruct and Alpaca data to collect covariance matrices in the instruction-previewed adaptation mode.

| Method | Context | MTBench |
|---|---|---|
| CorDA (ours) | WizardLM-Evol-Instruct | 5.15 |
| CorDA (ours) | Alpaca | 5.06 |

a rank of 128 for all linear layers in the model except the classification head. For our method, we sample train data from each of the tasks to initialize adapters in the instruction-previewed mode and fine-tune the corresponding task. As shown in Table 3, our method achieves the best performance on the MRPC, CoLA, and RTE tasks, and the highest average score.

## 4.4 Discussions

**Ablations.** We ablate the data choice used to produce covariance matrices and the adapter building manner in Table 4. The first row corresponds to the implementation of PiSSA that uses the largest $r$ singular values and their singular vectors as the initialized adapter by plain SVD. If we use the smallest $r$ components by plain SVD to build adapters, there is no apparent improvement in world knowledge benchmarks. This implies that the plain SVD cannot precisely capture world knowledge related ability into the principal components and thus freezing them does not help to mitigate knowledge forgetting. When our context-oriented decomposition (CO-SVD) is adopted with Wikitext-2, which is not closely correlated with question answering, the performance on world knowledge is much improved. When the context collected by the covariance matrix is from question answering data, *i.e.,* TriviaQA or NQ open, the world knowledge performance is further improved by a significant margin, and the average score is also enhanced as a result. Therefore, data context is important to orientate the decomposition process such that the characteristics of the ability concerned can be better aggregated into the principal components for maintaining or adapting. As shown in Table 5, similar to TriviaQA and NQ open in the knowledge-preserved adaptation, collecting context from different data sources belonging to the same category, namely WizardLM-Evol-Instruct and Alpaca, also results in close performance in the instruction-previewed adaptation. It is noteworthy that CorDA with Alpaca on MTBench (5.06) is still the highest among the compared baselines in Table 2.

**Limitations.** The two adaptation modes developed in this paper highlight different aspects in usage. However, the knowledge-preserved mode, while being adept at maintaining world knowledge, is naturally not advantageous on fine-tuning performance compared with the instruction-previewed mode. How to develop an initialization strategy [68] for adapters combining the merits of the two modes to maximize both objectives deserves future exploration.

## 5 Conclusion

In this paper, we propose a new parameter-efficient fine-tuning method, named context-oriented decomposition adaptation (CorDA). It performs singular value decomposition for pre-trained weights oriented by the covariance matrix that captures the context of the task concerned, and aggregates the context into the principal components for maintaining or adapting. Accordingly, our method is able to support two implementation modes, the knowledge-preserved adaptation to mitigate world knowledge forgetting and the instruction-previewed adaptation for better fine-tuning performance. In experiments, our knowledge-preserved adaptation not only achieves better fine-tuning performance than LoRA, but also maintains the world knowledge well, leading to the best average scores on three fine-tuning tasks. Our instruction-previewed adaptation is able to further enhance the fine-tuning performance, surpassing the state-of-the-art parameter-efficient fine-tuning methods DoRA and PiSSA.

## Acknowledgments and Disclosure of Funding

The research reported in this publication was supported by funding from King Abdullah University of Science and Technology (KAUST) - Center of Excellence for Generative AI, under award number 5940.

This work was also supported in part by the National Natural Science Foundation of China under Grant 62376069, in part by Young Elite Scientists Sponsorship Program by CAST under Grant 2023QNRC001, in part by Guangdong Basic and Applied Basic Research Foundation under Grant 2024A1515012027, and in part by the Shenzhen Science and Technology Program (Grant No. ZDSYS20230626091203008).

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

Table 6: The detailed numbers and more results of the experiment in Figure 2.

| Test Data | Method | discarded ranks | | | | | | | |
|---|---|---|---|---|---|---|---|---|---|
| | | 0 | 16 | 32 | 64 | 128 | 256 | 512 | 1024 |
| Wikitext-2 | Plain SVD | 5.47 | 6.32 | 7.31 | 9.89 | 18.03 | 18.5 | 25.42 | 73.92 |
| | ASVD [73] (with 256 Wiki samples) | 5.47 | 6.08 | 6.67 | 7.86 | 8.71 | 9.92 | 12.37 | 20.34 |
| | CO-SVD (with 32 Wiki samples) | 5.47 | 5.48 | 5.48 | 5.49 | 5.52 | 5.58 | 5.79 | 6.62 |
| | CO-SVD (with 256 Wiki samples) | 5.47 | 5.48 | 5.48 | 5.48 | 5.5 | 5.54 | 5.69 | **6.35** |
| | CO-SVD (with 256 PTB samples) | 5.47 | 5.49 | 5.5 | 5.52 | 5.57 | 5.74 | 6.25 | 8.69 |
| PTB | Plain SVD | 20.82 | 35.25 | 33.42 | 37.46 | 55.47 | 70.25 | 98.6 | 763.44 |
| | ASVD [73] (with 256 PTB samples) | 20.84 | 33.42 | 32.05 | 31.67 | 35.36 | 40.23 | 51.28 | 93.42 |
| | CO-SVD (with 32 PTB samples) | 20.75 | 20.75 | 20.76 | 20.78 | 20.83 | 20.91 | 21.17 | 22.68 |
| | CO-SVD (with 256 PTB samples) | 20.88 | 20.88 | 20.88 | 20.89 | 20.91 | 20.94 | 21.14 | **22.28** |
| | CO-SVD (with 256 Wiki samples) | 20.34 | 20.34 | 20.32 | 20.41 | 20.59 | 21.25 | 22.94 | 29.69 |

# A    Appendix: Implementation Details

## A.1    Fientuning on Math, Code, and Instruction Following

For fine-tuning tasks on Math, Code, and Instruction Following, we adopt the same training setting as PiSSA [47]. Concretely, optimization is performed with the AdamW optimizer, a batch size of 128, and a learning rate of 2e-5. We employ cosine annealing schedules with a warmup ratio of 0.03 and do not apply weight decay. Training is conducted exclusively on the first 100,000 conversations from the dataset for one epoch, with loss computation solely based on the response. Our experiments are executed on the NVIDIA A100-SXM4(40/80GB) GPUs. Publicly available platforms are utilized for the evaluation of world knowledge (TriviaQA, NQ open, and Web QS) [2], Code (HumanEval and MBPP) [3], and Instruction Following (MTBench) [4].

## A.2    GLUE Benchmark

To ensure fair comparison across Full fine-tuning, LoRA, DoRA, and CorDA in the GLUE benchmark, we implement all methods under the same training and evaluation settings. The AdamW optimizer is used with a batch size of 32 and a learning rate of 4e-5 for 3 epochs, following a linear learning rate schedule. The max token length is set as 128. The rank of LoRA, DoRA, and our CorDA is 128. For covariance matrix collection of CorDA, we concatenate the representative content of each training sample to form a text sequence. From this sequence, 256 text segments, each containing 256 tokens, are randomly sampled. The selected content for each task is as follows: MRPC, RTE, and STS-B using "sentence1", CoLA and SST-2 using "sentence", and QNLI using "question". All methods are trained on a single NVIDIA A100-SXM4(40/80GB) GPU.

# B    Appendix: More Results

Table 6 lists the detailed numbers and more results of the experiment in Figure 2. It is shown that the number of sampled data only has a very limited impact. When the smallest 1024 ranks are discarded, using 32 samples is slightly worse than 256 samples in both Wikitext-2 and PTB. It implies that a small number of samples is enough to capture context into the principal components. Besides, collecting samples from the same dataset as the one used to test is able to attain a better performance after discarding a large number of ranks. For example, when discarding the smallest 1024 ranks, CO-SVD (with 256 Wiki samples) is better than CO-SVD (with 256 PTB samples) on Wikitext-2 (6.35 v.s. 8.69), and CO-SVD (with 256 PTB samples) is better than CO-SVD (with 256 Wiki samples) on PTB (22.28 v.s. 29.69). This also reveals that precisely capturing the data context in our decomposition is crucial for better maintaining the task characteristics into the principal components, and explains why our method is superior to the PEFT methods without considering data context.

---

[2] https://github.com/EleutherAI/lm-evaluation-harness
[3] https://github.com/bigcode-project/bigcode-evaluation-harness
[4] https://github.com/lm-sys/FastChat

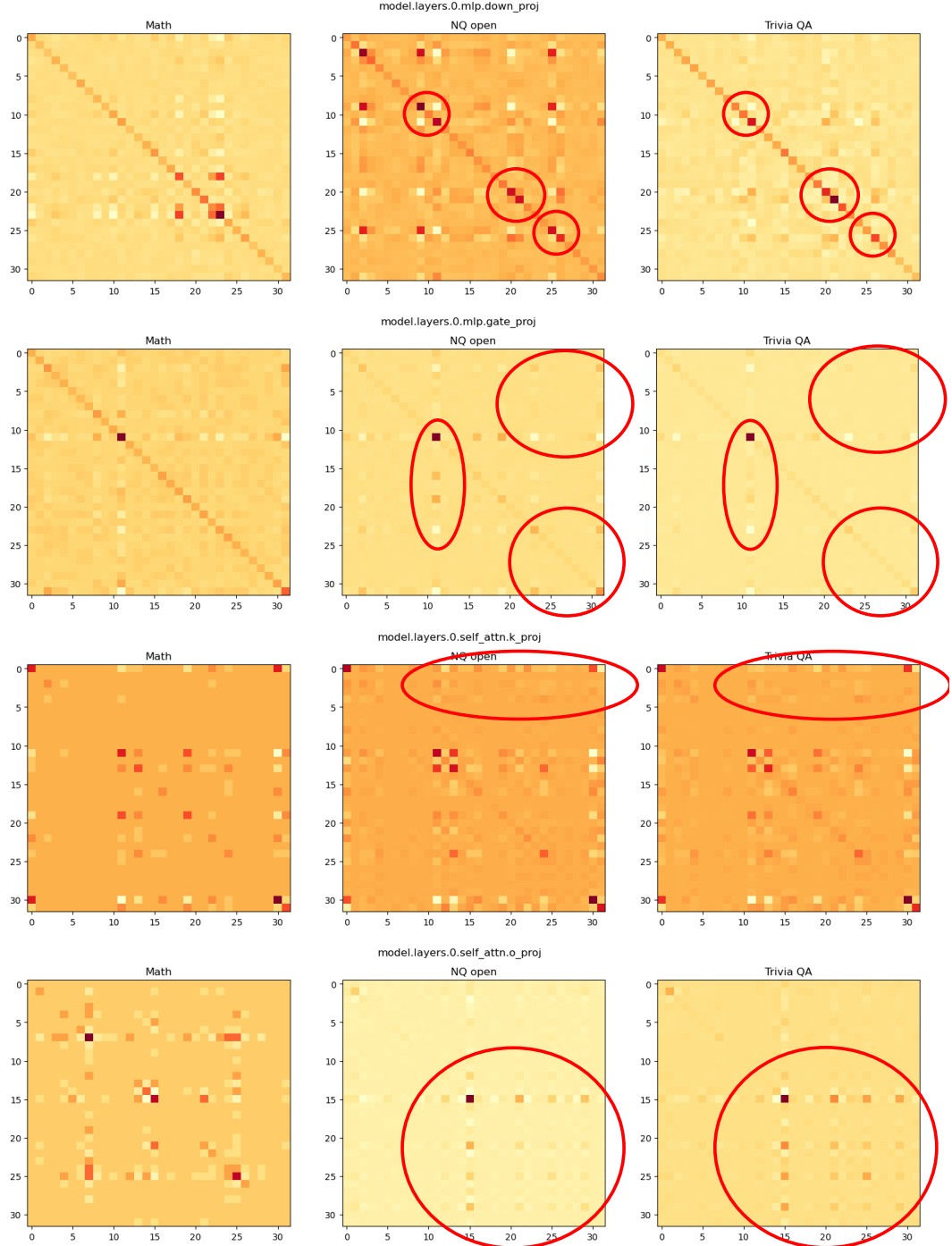

Figure 4: Covariance visualization results for "self_attn.k_proj", "self_attn.o_proj", "mlp.down_proj", and "mlp.gate_proj" weights in the 0-th layer. Please zoom in for a better view.

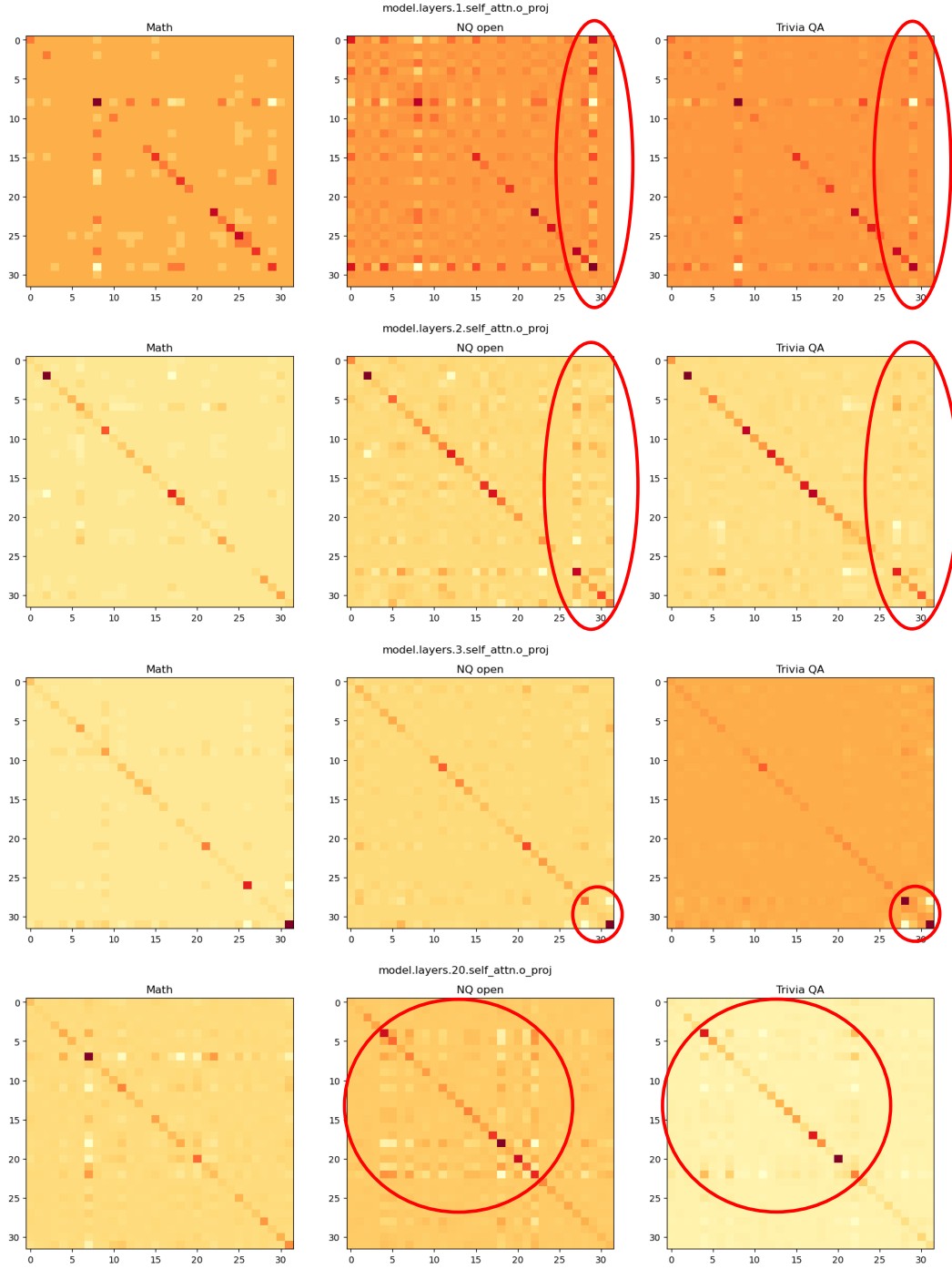

Figure 5: Covariance visualization results for "self_attn.o_proj" weights in different depth layers. Please zoom in for a better view.

# C  Appendix: Covariance Visualization Results

We provide the visualization results of the covariance matrices collected from three tasks MetaMath, NQ open, and Trivia QA in Figures 4 and 5.

Since the original dimension in 4096 or 11008 will be too large to be informative, we down-sample the covariance matrices into $32 \times 32$ and visualize their heatmaps. We provide the results from the activations before different linear weights including `self_attn.k_proj` (the same as `self_attn.q_proj` and `self_attn.v_proj` due to the same input), `self_attn.o_proj`, `mlp.down_proj`, and `mlp.gate_proj` (the same as `mlp.up_proj`) in the first layer, and the `self_attn.o_proj` weight in later layers. It is shown that the heatmaps from NQopen and TriviaQA (both are QA tasks) share some similar patterns (marked in red circles), which do not appear in the heatmap from the different task MetaMath. Therefore, when the inputs of different tasks are fed into an LLM, the covariance matrix from activations will exhibit different patterns. The visualization result empirically supports that the covariance matrix patterns can be used to characterize the triggered task. We use such patterns to orientate the decomposition of LLM pretrained weights, to make the resulting adapter initialization task-dependent.

