# OpenReview forum: "CorDA: Context-Oriented Decomposition Adaptation of Large Language Models for Task-Aware Parameter-Efficient Fine-tuning"
_NeurIPS.cc/2024/Conference — NeurIPS 2024 poster_

### Official Review · Reviewer_kduk · 2024-06-22

**Soundness:** 3
**Presentation:** 3
**Contribution:** 2
**Rating:** 6
**Confidence:** 4

**Summary:**

This paper proposes CorDA, a context-oriented decomposition adaptation method that initializes LoRA adapter with different components from the weight decomposition to support two different options: knowledge-preserved adaptation and instruction-previewed adaptions. Through extensive experiments on LLaMA-2-7b and RoBERTa, the paper shows that the knowledge-preserved adaptation can better preserve performance on general knowledge task and the instruction-previewed adaptations can yield better target task performance compared to the baseline PEFT methods.

**Strengths:**

1. The experimental results on the instruction-previewed mode are strong. Table 2 and Table 3 demonstrate that the purposed method yields better performance when fine-tuning on targets compared with other PEFT methods and its performance is nearly on par with full finetuning.
2. The proposed method is intuitive and simple in concept. If the decoupled components capture context in the principle components decoupled, the two modes are well-motivated.

**Weaknesses:**

1. One major concern of this work is the lack of theoretical support, and some arguments in the paper are not well justified.
    - Line 45-47: “The covariance matrix of each layer’s activation … are responsive to the task triggered to highlight different aspects of the pre-train weight.” Is there any support for this or is this a pure intuition?
    - Why do you perform SVD on $WC$ ? What does $WC$ represent?
2. The knowledge-preserved mode is not very compelling given this method will hurt the target task performance and cannot achieve the best of both worlds by design. Although the authors have already recognized this when stating the limitations, I don’t know why the paper emphasizes this mode in parallel with the instruction-previewed adaptation mode. Regarding the design of method to achieve the best of both worlds, [1] may be a pointer for reference as it also studies how to preserve the forgetting of general knowledge in LMs when continual training them.

[1] Continual Pre-training of Language Models, Ke et al., ICLR 2023

**Update:** Most of points are addressed by the author response, so I update my score from 4 to 6.

**Questions:**

1. Does the major difference between CorDA and LoRA lie in the initialization? Have you compared with other PEFT methods that focuses on initialization technique? Currently, the baseline comparison is not very comprehensive.

**Limitations:**

Yes, it's discussed from Line 293.

---

> ### Author Rebuttal · Authors · 2024-08-06
>
> Thank you for recognizing the contributions of our work and your valuable comments.
>
> > (1) One major concern is the lack of theoretical support, and some arguments are not well justified. Is there any support for Line 45-47 or is this a pure intuition? Why SVD on $WC$? What does $WC$ represent?
>
> We first highlight the significance of our work even though it is lacking a theoretical support, and then provide some supports from literature and our empirical analysis justifing Line 45-47. Finally, we explain our method of SVD on $WC$.
>
> _1. Significance of our work even with no theory:_
>
> Despite the lack of theory, the contribution of our work is significant for the following reasons
>
> - **It is not necessary to use a theory to show its validity.** Activation awareness has been proven to be important in LLM compression and quantization [27, 35, 65]. But in PEFT, existing studies rarely consider task context.
> So, a fundamental originality of our method is that we introduce task context such that the resulting adapter initialization is task-dependent. We provide some other supports (in _"2. Supports for Line 45-47"_) for why using covariance matrix and its advantage over only using the activation itself.
>
> - Existing _PEFT methods_ rarely support the option of finetuning with the pre-trained knowledge better preserved (**[1] is not a PEFT method**). Since our method is task-dependent, we offer such flexibility enabling both knowledge-preserved adaptation and instruction-previewed adaptation, customized for the actual need. When knowledge maintenance is a need, the former satisfies this demand. When we only want to push the limit of the downstream tasks without concerning about the pre-trained knowledge maintenance, the latter is favorable.
>
> - In experiments, compared with LoRA, our knowledge-preserved adaptation achieves improvements on both worlds (shown in Table 1), enjoying better results than LoRA on both world knowledge benchmarks and downstream tasks, and having the best comprehensive average performance among all the compared methods.
> Our instruction-previewed adaptation is able to further improve the downstream task performance, surpassing LoRA, DoRA, and PiSSA in all the three finetuning tasks (shown in Table 2). In either case, the experimental performance is a progress in the field.
>
> Therefore, we believe that our contributions should not be unappreciated just because our method lacks a theoretical support, which is actually a common issue in existing PEFT methods. Actually, there is no theorem in the original LoRA paper and most of the follow-up studies.
>
> _2. Supports for Line 45-47:_
>
> - **Support from literature**. Activation awareness has been proven to be effective in LLM compression and quantization [27, 35, 65]. OWQ [27] proposes to leverage the outlier awareness from covariance matrix to make weight quantization task-specific. ASVD [65] also considers outliers, scaling the pre-trained weight with the input activation to perform SVD for model compression. The rationality behind these studies is that the outlier in activation or covariance matrix is responsive to the task triggered by the input, which is also our motivation of using covariance matrix to orientate the decomposition.
>
> - **Support from empirical analysis**. As suggested by Reviewer Mgdn, we visualize the covariance matrix collected from samples from the three tasks, MetaMath, NQ open, and Trivia QA. Note that we downsample the covariance matrices from a high dimension (4096 or 11008) into $32 \times 32$, and visualize their heatmaps. Please see the **PDF file of the global rebuttal** for the visualization results. It is shown that the heatmaps from NQopen and TriviaQA (both are QA tasks) share some similar patterns marked in red circles, which do not appear in the heatmap from the different task MetaMath. The visualization result further supports that covariance matrix can be used to characterize the triggered task.
>
> - **Support from experimental results**. As stated above, ASVD and our method has a similar motivation. ASVD performs SVD on the pre-trained weights scaled by input activation for model compression. We use covariance matrix to orientate the decomposition of weights for building adapters. We compare our context-oriented SVD (CO-SVD) with ASVD in Figure 2 and Table 6 of our paper. It is shown that when discarding the smallest several components, CO-SVD is much better at maintaining the performance of WikiText-2 and PTB than ASVD and Plain SVD. This result indicates that our CO-SVD has a stronger ability in assembling task context into its principle components. Therefore, covariance matrix is better choice for us to build task-dependent adapters.
>
> _3. Why SVD on $WC$ and what does $WC$ represent:_
>
> As we respond above, our CO-SVD, using covariance matrix to orientate the decomposition of pre-trained weights, has a strong ability in assembling task context into the principle components. As shown in Figure 2 and Table 6, our CO-SVD, _i.e._ SVD on $WC$, is much better than the plain SVD that does not include task context, and ASVD that performs SVD on the weights scaled by activation.
> So, we perform SVD on $WC$, where $W$ is the pre-trained weight and $C$ is the covariance matrix collected from a few samples. Its principle components after decomposition are task dependent and better capture task context than only using the activation itself.
>
>
> > (2) The knowledge-preserved mode is not very compelling given this method will hurt the target task performance. Why the paper emphasizes this mode in parallel with the instruction-previewed adaptation mode.
> >
> > (3) The other questinos
>
> We have discussion about the unique value of our knowledge-preserved mode and answer to your remaining questions in the following comment.

---

> ### Author Response · Authors · 2024-08-07
> **Discussion about the value of knowledge-preserved mode and answers to remaining questions**
>
> > (2) The knowledge-preserved mode is not very compelling given this method will hurt the target task performance. I don’t know why the paper emphasizes this mode in parallel with the instruction-previewed adaptation mode. [1] may be a pointer for reference as it also studies how to preserve the forgetting of general knowledge in LMs when continual training them.
>
> **Please note that the knowledge-preserved adaptation does NOT hurt the target task performance**. As shown in Table 1, compared with LoRA, our method has better performance than LoRA in both worlds (world knowledge benchmarks and downstream tasks) in most cases. We **quote some results of LoRA and CorDA in Table 1 here** for your reference.
>
> |Method|Trivia QA | NQ open | WebQS | GSM8k | Math | Avg. |
> |---|---|---|---|---|---|---|
> |LoRA|44.17 | 1.91 |6.64| 42.68| 5.92| 20.26|
> |CorDA | 44.30 | 9.36 | 7.14 | 44.58 | 6.92 | 22.46 |
>
> |Method|Trivia QA | NQ open | WebQS | MTBench | Avg.|
> |---|---|---|---|---|---|
> |LoRA|47.46 |10.28| 7.73| 4.60| 17.52|
> |CorDA | 50.34 |14.43| 8.17| 5.05| 19.50|
>
> More importantly, as we respond above, existing PEFT methods rarely consider or support finetuning with knowledge better perserved. There are some studies on the continual training of LLMs [1] and [14, 21] (ref. in our paper), but they are not PEFT methods. We will cite [1] in the revised version of our paper.
> As shown in Table 1, when using the average performance of the both worlds to measure the comprehensive ability, CorDA in knowledge-preserved adaptation achieves the best results among all the compared methods in all the three tasks. **Therefore, we respectfully do not agree with your comment that "knowledge-preserved adaptation is not compelling" and "it hurts the target task performance".**
>
> Admittedly, CorDA in knowledge-preserved adaptation is not stronger than the instruction-previewed adaptation when only evaluating the downstream task performance. But please note that maintaining pre-trained knowledge better and pursuing a better finetuning performance is inherently a tradeoff (also mentioned in [14] and [21]). A method may be better than another one in both worlds, just like CorDA v.s. LoRA in the results above, but for the method itself, the two worlds are still a tradeoff. We would like to use an analogy to support. The advanced neural architecture (_e.g._ Transformer) may be better than a traditional MLP/CNN architecture in terms of both accuracy and parameter efficiency. But for the better architecture itself, higher accuracy still brings more parameters (BERT_large is better than BERT_base with more parameters).
>
> Therefore, when knowledge maintenance is not a concern, we introduce our instruction-previewed adaptation, which puts all its efforts for the downstream task, surpassing the competitive studies DoRA and PiSSA in the three finetuning tasks, Math, Code, and instruction following, as shown in Table 2.
> In conclusion, the knowledge-preserved adaptation and instruction-previewed adaptation highlight the comprehensive performance and the specialized ability of downstream task, respectively. We think this is also a feature of our PEFT method, allowing for customized selectivity based on the actual need.
> That is why we emphasize the two modes in parallel.
>
> [1] Continual Pre-training of Language Models, Ke et al., ICLR 2023
>
>
> > (3) Questions: Does the major difference between CorDA and LoRA lie in the initialization? Have you compared with other PEFT methods that focuses on initialization technique? Currently, the baseline comparison is not very comprehensive.
>
> Yes, CorDA brings task context into the LoRA adapter initialization.
> Adopting the same LoRA structure not only facilitates fair comparison, but also enables to restore the orignal LLM architecture after finetuning without architectural change or introducing inference burden.
> Yes, we have compared with PiSSA, which also focues on the LoRA adapter initialization but does not consider task context.
> DoRA builds the adapter with a normalization and a learnable magnitude, and also does not consider task context.
> It is noteworthy that both DoRA (ICML 24) and PiSSA (released on Arxiv in April 2024) are recent studies and are strong baselines. Besides, full parameter finetuning is the most direct reference because it usually has the best finetuning performance without considering parameter efficiency.
> For downstream tasks, as shown in Table 2 and Table 3, our method achieves finetuning performances on par with full parameter finetuning, and better performances than the compared PEFT methods LoRA, DoRA, and PiSSA.
> For comprehensive ability (with knowledge benchmarks included), as shown in Table 1, our method has the best average performance among full finetuning and the PEFT methods.
> Therefore, our experimental results are already able to demonstrate the effectiveness of our proposed method.

---

> > ### Comment · Reviewer_kduk · 2024-08-10
> >
> > I thank the authors for the detailed response.
> >
> > The first part of the response adequately addresses my Weakness 1. However, it's worth noticing that I am not saying every paper necessarily needs to have a theoretical part. Solid, inspiring empirical results are very important for AI/ML application. The reason why I raised Weakness 1 is the current writing of Section 3.2 gives people an impression that the paper states that the covariance matrix is tightly connected with task context/patterns, where the latter concept itself does not have a formal definition (it's really hard to articulate what task context is). I suggest weakening the argument here and discussing the empirical comparison more, as it's the empirical results that show decomposing $WC$ could lead to a better results.
> >
> > Overall, the author response addresses my major concern about the method proposed in this paper, so I increase my score accordingly.

---

> > > ### Author Response · Authors · 2024-08-10
> > >
> > > Dear Reviewer kduk,
> > >
> > > Thank you for increasing the score and your suggestions. We will rephrase the argument, and include more discussion about the empirical comparison in the revised paper.
> > >
> > > Authors

---

### Official Review · Reviewer_HCBt · 2024-07-09

**Soundness:** 3
**Presentation:** 2
**Contribution:** 3
**Rating:** 7
**Confidence:** 3

**Summary:**

The paper proposes a context-oriented decomposition adaptation method for large language models (LLMs) called CorDA. This method constructs learnable adapters by considering the context of downstream tasks or world knowledge. It can bridge the performance gap between parameter-efficient fine-tuning (PEFT) and full-parameter fine-tuning, while also mitigate the problem of catastrophic forgetting. Specifically, the method decomposes the weights of LLMs using singular value decomposition (SVD) guided by the covariance matrix of input activations. Through this approach, the model can identify important weights related to specific contexts, thereby creating context-aware adapters. These adapters can be customized for different tasks, enabling the model to retain general knowledge while enhancing performance on specific tasks.

**Strengths:**

S1: The structure of the paper is relatively clear. The introduction progressively leads readers to understand the current state of fine-tuning LLMs and the challenges posed by knowledge forgetting after fine-tuning. This approach allows readers to quickly grasp the current situation of LLMs fine-tuning. In the subsequent method section, related modules are introduced based on these challenges.
S2:The main challenge addressed in this paper is the catastrophic forgetting of knowledge when fine-tuning LLMs. To tackle this, the authors propose a context-oriented decomposition method and introduce two modules. The first module generates an adapter that retains the model’s general knowledge by activating the covariance matrix using other question-answering datasets. The second module, designed to adapt to task-specific instructions, is called the instruction preview adaptation module, which generates adapters specific to the given tasks. The paper thoroughly explains the implementation process of the method, including the calculation of the covariance matrix and the application of SVD.
S3:The experimental validation in this paper is comprehensive, verifying the effectiveness of each module. The results of the modules are thoroughly analyzed through extensive experiments, demonstrating the effectiveness of the proposed modules.

**Weaknesses:**

S1:In the experimental section, it is necessary to add a description of the experimental environment and provide parameters to offer readers more possibilities for replication and ensure authenticity. Additionally, the paper lacks specific quantification of catastrophic forgetting in the model.

**Questions:**

none

---

> ### Author Rebuttal · Authors · 2024-08-06
>
> Thank you for recognizing the contributions of our work and your valuable comments.
>
> > (1) It is necessary to add a description of the experimental environment and provide parameters to offer readers more possibilities for replication and ensure authenticity. Lacks specific quantification of catastrophic forgetting in the model.
>
> We have provided the implementation details in our main paper and Appendix A. Concretely, we describe the data choice and number to collect covariance matrix, the rank choice, and datasets used in the experiment section of our paper. In Appendix A, we provide the training settings (optimizer, batchsize, learning rate, etc), the GPU device we use, and the evaluation tools. Our code and models will be publicly available. Please let us know if the reviewer has any question about our experimental details.
>
> Usually the average performance and the performance drop compared with the model before training/finetuning are the common quantification metrics to evaluatuate catastrophic forgetting. We have shown the average performance over world knowledge benchmarks and downstream tasks in Table 1.
> It can reflect the comprehensive ability of the finetuned model in maintaining the pre-trained knowledge while learning the new task.
> Besides, for the world knowledge benchmarks (Trivia QA, NQ open, and WebQS columns), the performance difference between the second row (LLaMA-2-7B) and the rows below shows the performance drops of each finetuning method compared with the original LLaMA-2-7B model. It is shown that our method enjoys the lowest performance drop in most cases. We will mark the performance drop beside these numbers to show the advantage more explicitly.

---

> > ### Comment · Reviewer_HCBt · 2024-08-13
> >
> > Thanks for your responses. After going through these responses,  I decide to manintain my score.

---

### Official Review · Reviewer_Mgdn · 2024-07-13

**Soundness:** 3
**Presentation:** 2
**Contribution:** 3
**Rating:** 6
**Confidence:** 3

**Summary:**

This paper introduces a new parameter-efficient fine-tuning (peft) method called Context-oriented Decomposition Adaptation (CorDA). Although fundamentally similar to LoRA, it differs in that it initializes two low-rank matrices using the SVD results of pre-trained weights reflecting the data context. To incorporate the data context, few samples specific to the task are required, utilizing the input activations of these samples for the large language model (LLM). CorDA can be applied in two ways: knowledge-preserved adaptation and instruction-previewed adaptation. The former aims to retain specific knowledge during fine-tuning, while the latter is used to adapt the LLM more effectively to the desired instructions.

**Strengths:**

- **Originality**: The approach of modifying the LoRA method to maintain or enhance the performance of language models on specific data types appears to be a novel attempt. This method has not been explored in existing works dealing with parameter-efficient fine-tuning, thus offering originality.
- **Quality**: Experimental results demonstrate that CorDA is more effective than existing methods like LoRA or PiSSA in fine-tuning models such as Llama and RoBERTa.
- **Significance**: The idea of using data to derive effective low-rank matrices could inspire future research.

**Weaknesses:**

- **Originality**: There are methodological similarities with PiSSA, and comparisons with existing works like AdaLoRA [1] are lacking.
- **Quality**: The exact role of the covariance matrix $C$ is not clearly explained. The paper states in lines 145-147 that “the covariance matrix of each layer’s activation will exhibit different outlier patterns as they are responsive to the task triggered to highlight **different aspects** of the pre-trained weight.” Providing visualizations to show how data from different tasks trigger different parts of the weight would enhance clarity.
- **Clarity**: The amount and type of data used to derive the covariance matrix $C$ significantly impact performance, but the paper lacks a clear discussion and analysis on this. Moreover, there is no guidance on how to choose $r$ in Equations 4 and 5, which appears to be a crucial hyperparameter. This omission makes it challenging to practically apply the proposed method based solely on the information provided in the paper.
- **Significance**: Although the research seems advantageous compared to existing parameter-efficient fine-tuning methods, it is unclear how much and what quality of data is needed to derive $C$ for better performance. The usability of the CorDA heavily depends on the amount and quality of data required. Despite this, the concept of using data to derive better low-rank matrices is novel.

Reference

[1] AdaLoRA: Adaptive Budget Allocation For Parameter-efficient Fine-tuning

**Questions:**

- It would be beneficial to include visualizations of the covariance matrix derived from different data.
- The paper should incorporate a data selection strategy for $C$ and an extensive analysis on $r$.

**Limitations:**

The authors adequately addressed the limitations in Section 4.4.

---

> ### Author Rebuttal · Authors · 2024-08-06
>
> Thank you for recognizing the contributions of our work and your valuable comments.
>
> > (1) Originality: methodological similarities with PiSSA, and comparisons with existing works like AdaLoRA
>
> _Similarity with PiSSA:_
>
> Both our method and PiSSA adopt SVD for the pre-trained weights, however, our method has fundamental difference and advantage compared with PiSSA as follows:
>
> - Activation awareness has been proven to be important in LLM compression and quantization. But in PEFT, existing studies rarely consider task context. PiSSA provides a better adapter initialization than LoRA using SVD, but still does not consider any task context. **Although our method also adopts SVD, we never claim that the novelty of our method lies in the usage of SVD to build LoRA adapters. The fundamental originality is that we introduce task context, captured by the covariance matrix from activations, to orientate the decomposition of weights, such that the resulting adapter initialization is task-dependent**.
>
> - Existing PEFT methods including PiSSA rarely support the option of finetuning with pre-trained knowledge better preserved. Since our method is task-dependent, we offer such flexibility enabling both knowledge-preserved adaptation and instruction-previewed adaptation, customized for the actual need.
>
> - Even when the maintenance of pre-trained knowledge is not a concern, our method in instruction-previewed adaptation surpasses DoRA and PiSSA in the three downstream tasks of Math, Coding, and instruction following, as shown in Table 2 of our paper. It is noteworthy that both DoRA (ICML24) and PiSSA (arXiv:2404.02948, online in April 2024, within 2 months of our submission) are the latest studies, and PiSSA actually could be considered as contemporaneous.
>
> Therefore, we believe that **our originality and contributions, including the methodology of introducing task context into LoRA adapter initialization, and the experimental performance of both maintaining pre-trained world knowledge and improving downstream task ability, should NOT be unappreciated just because we also adopt SVD in the adapter initialization process**.
>
> _Comparison with AdaLoRA:_
>
> Our method improves the LoRA adapter initialization by introducing task context, and adopts the low intrinsic dimension $r$ following the standard setting in LoRA.
> AdaLoRA aims to dynamically adjust $r$ during finetuning. So, CorDA and AdaLoRA are methods focusing on different aspects (how to better build adapters and how to dynamically adjust the rank).
> In our experiments, we mainly compare CorDA with methods of the same stream, PiSSA and DoRA. PiSSA also focuses on the LoRA adapter initialization using SVD, but does not consider task context. DoRA builds the adapter with a normalization and a learnable magnitude, and also does not consider task context. It is noteworthy that both DoRA (ICML 24) and PiSSA (released on Arxiv in April 2024) are recent studies and are strong baselines. Besides, full parameter finetuning is the most direct reference because it usually has the best finetuning performance without considering parameter efficiency. For downstream tasks, as shown in Table 2 and Table 3, our method achieves finetuning performances on par with full parameter finetuning, and better performances than the compared PEFT methods LoRA, DoRA, and PiSSA. For comprehensive ability (with knowledge benchmarks included), as shown in Table 1, our method has the best average performance among full parameter finetuning and the PEFT methods. Therefore, our experimental results are already able to demonstrate the effectiveness of our proposed method.
>
> > (2) Quality: The exact role of the covariance matrix $C$ is not clearly explained. The paper states that “the covariance matrix of each layer’s activation will exhibit different outlier patterns as they are responsive to the task triggered to highlight different aspects of the pre-trained weight.” Providing visualizations to show how data from different tasks trigger different parts of the weight would enhance clarity.
>
> When the inputs of different tasks are fed into an LLM, the covariance matrix from activations will exhibit different patterns. We use such patterns to orientate the decomposition of LLM pretrained weights, to make the resulting adapter initialization task-dependent.
> We did not provide the visualization of the covariance matrix because the dimension in 4096 or 11008 is too large to be informative. In the rebuttal, we downsample the covariance matrices into 32 $\times$ 32 and visualize their heatmaps.
> Please refer to our response in the **global rebuttal and its PDF attached**.
> It is shown that the heatmaps from NQopen and TriviaQA share some similar patterns (marked in red circles), which do not appear in the one from the different task MetaMath.
> We hope this result can further justify that the covariance matrix patterns can be used to characterize the triggered task.
>
> Besides, we can find some supports from the literature. OWQ [27] proposes outlier-aware weight quantization based on covariance matrix. ASVD [65] performs SVD considering activations for compression. But as shown in Figure 2 and Table 6, our CO-SVD based on covariance matrix instead of only the activation itself, is much better at capturing the task context.
>
> We will add these visualization results and more discussion about the literature support in the revised version of our paper.
>
> > (3) Clarity and Significance: "The amount and type of data used to derive the covariance matrix $C$ significantly impact performance, but the paper lacks a clear discussion and analysis on this. Moreover, there is no guidance on how to choose $r$ in Equations 4 and 5, which appears to be a crucial hyperparameter."
>
> **We address your concern about the impact of data amount and type to derive $C$ on performance and how to choose $r$ in the global rebuttal. Please refer to the global rebuttal-> common issue.**

---

> > ### Comment · Reviewer_Mgdn · 2024-08-11
> >
> > Thank the author for the clarifications. The response effectively addresses my concerns, clearly highlighting the key contributions of this work in comparison to PiSSA, as well as the different patterns of covariance matrices across tasks. Therefore, I have increased the score accordingly.

---

### Official Review · Reviewer_527n · 2024-07-14

**Soundness:** 4
**Presentation:** 4
**Contribution:** 3
**Rating:** 7
**Confidence:** 4

**Summary:**

The paper proposes an initializes algorithm for LoRA based fine-tuning, with the aim to maintain world knowledge and also improve training performance on the fine-tuning task at hand. The authors carefully explain the reasoning behind their initialization scheme, and conduct extensive studies to showcase the efficacy of their framework. Furthermore, they show that they can get surprising improvements on MT bench, simply by maintaining knowledge information. Overall, the paper presents a systematic approach to initializing LoRA parameters, and the general principle can guide the community towards better training of large language models.

**Strengths:**

The strength of the paper lies in its simplistic exposition of its motivation, proposed framework, and the extensive experimental study to show the efficacy of the initialization framework. The authors begin with a clean exposition to SVD, and how one can think of the different eigenvectors in the input weight covariance matrix. With a very intuitive discussion, the authors show the directions along which the weight parameters aren't perturbed during training to maintain world knowledge. The directions are selected by looking at the covariance matrix of  knowledge-test datasets like TriviaQA, and NaturalQA. Furthermore, the authors indicate the directions along which they want to pre-capture features of the fine-tuning task at hand, which can give bigger returns compared to training LoRA parameters from scratch. Overall, the principle of data-dependent initialization is novel and can guide the community towards more systematic training designs.

**Weaknesses:**

Overall, I don't see much weakness with the work. I have a few questions regarding the experimental framework, which I would like to discuss during the rebuttal period.


- How did the authors decide to maintain the last $r$ eigenvectors for world-knowledge? I believe, the authors should cite relevant works or make proper justification on this design choice.

- How scalable is their proposed approach to other datasets where we may want to maintain knowledge? That is, how likely is the model to maintain knowledge on TriviaQA, if it wasn't included during LoRA initialization?

- What is the necessary sample complexity for initializing the LoRA parameters, i.e. how many samples from the knowledge datasets are necessary to get a good estimate of the directions to freeze?

- Is the knowledge preservation necessary at each weight parameters and each layer of the model? How do results change when the model's LoRA parameters are restricted only in the lower layers, while the higher layers are given more freedom during fine-tuning?

**Questions:**

Please check my questions above.

**Limitations:**

The authors discuss the limitations of their work in section 4.4, and clearly indicate the future directions that the community can pursue starting from their work.

---

> ### Author Rebuttal · Authors · 2024-08-06
>
> Thank you for recognizing the contributions of our work and your valuable comments.
>
> > (1) How did the authors decide to maintain the last $r$ eigenvectors for world-knowledge? Cite relevant works or make proper justification on this design choice.
>
> As shown in Figure 1 and Eq. (4), in the knowledge preserved adaptation, we use the last $r$ eigenvectors to build adapters, instead of maintaining them. We explain the design choice in more details as follows.
>
> Our context-oriented SVD (CO-SVD) decomposes the pre-trained weights into orthogonal abilities where the largest principle components most correspond to the task context captured by the covariance matrix. In instruction previewed adaptation (IPA), we want to better learn the target ability, so we use the first $r$ eigenvectors to build adapters. In knowledge preserved adaptation (KPA), the first components correspond to the QA ability, which is what we want to maintain. So, we use the last $r$ eigenvectors to build adapters while freezing the first ones.
> Therefore, the design motivations of IPA and KPA are not opposite. The principle components after CO-SVD both represent the ability indicated by the covariance matrix. The difference is a result of the purpose, _i.e._ better adapt these components (used as adapters in IPA) or better maintain these components (frozen in KPA).
>
> We thank the reviewer for the suggestion of citing relevant works to justify the design choice. Actually, ASVD [65] is just based on a similar design motivation. They discard the last eigenvectors and only maintain the largest several components for model compression. In our KPA, we also maintain the principle components to preserve world knowledge, but adapt the last $r$ eigenvectors to learn new ability instead. Despite a similar motivation, we have shown that our CO-SVD is much better at capturing the characteristics of the target task than ASVD in Figure 2 and Table 6. Previous stuides [A, B, C] also adopt SVD for model compression. We will cite these works and explain the design choice of KPA and IPA more thoroughly in the revised paper.
>
> [A] Language model compression with weighted low-rank factorization, ICLR'22.
>
> [B] Exploiting Linear Structure Within Convolutional Networks for Efficient Evaluation, NeurIPS'14.
>
> [C] GroupReduce: Block-Wise Low-Rank Approximation for Neural Language Model Shrinking, NeurIPS'18.
>
> > (2) How scalable is their proposed approach to other datasets where we may want to maintain knowledge? How likely is the model to maintain knowledge on TriviaQA, if it wasn't included during LoRA initialization?
>
> Our method guides the decomposition of pretrained weights by the data-dependent covariance matrices. However, the knowledge to preserve is definitely **NOT** constrained to the data used for adapter initialization. Actually, we only sample 256 questions from a QA dataset to collect the covariance matrices. The ability of our method to preserve world knowledge cannot be derived from these limited samples themselves. That is to say, what we maintain is the QA ability, instead of some specific knowledge from the collected data.
>
> We use the covariance matrices, whose outlier patterns characterize the target task, to make the decomposition task-specific. Therefore, the same kinds of input/query (e.g. questions from TriviaQA and NQopen) have a similar effect because they both trigger a similar ability.
> We have conducted analysis in Table 4 (TriviaQA and NQopen) and Table 5 (WizardLM-Evol-Instruct and Alpaca) comparing results with data from similar tasks.
> Please refer to our response in **the global rebuttal -> common issue -> (2) Type/quality.**
> These results indicate that randomly collecting context from one dataset has a scalable effect to other datasets of the same task.
>
> > (3) What is the necessary sample complexity for initializing the LoRA parameters, i.e. how many samples from the knowledge datasets are necessary to get a good estimate of the directions to freeze?
>
> As described in Line 233 of our paper, we sample 256 questions for our experiments of knowledge preserved adaptation.
> We collect 256 samples in all our experiments for both KPA and IPA modes.
>
> In Table 6, we analyze the effect of sample number. We compare the results of collecting 32 and 256 samples for Wikitext-2 and PTB, respectively. In order to also investigate the effect of sample number in the KPA experiment, we collect less samples (128 and 32) from NQopen. Please refer to our response and result in **the global rebuttal -> common issue -> (1) Amount.**
>
> > (4) Is knowledge preservation necessary at each weight parameters and each layer? How do results change when the model's LoRA parameters are restricted only in the lower layers, while the higher layers are given more freedom?
>
> We follow the standard setting in LoRA, DoRA, and PiSSA, _i.e._, evenly using the same low rank for all linear layers. It also facilitates fair comparison with them.
> Since large deviation of lower layer weights will go through more later layers whose accumulative effect will cause a large shift for the final representation, restricting the adaptation of lower layers and finetuning higher layers with more freedom indeed may be preferable to maintain the pre-trained knowledge.
> Thus we can use a small intrinsic rank ($r$) for lower layers to constrain their adaptation, and adopt a large $r$ and even full finetuning for higher layers to learn downstream tasks.
> We can also adopt an adaptive strategy, _e.g._ based on the eigenvalue distribution after our CO-SVD, to assign ranks.
> These strategies may be more parameter-efficient than using the same rank for all layers.
> Besides, Transformer different weights and MLP blocks may also play different roles in maintaining pre-trained knowledge. Investigating the importance of different weight parameters in preserving knowledge and assigning adapters accordingly will be a valuable extension of CorDA and deserve our future exploration.

---

> > ### Comment · Reviewer_527n · 2024-08-12
> >
> > I thank the authors for their response. After going through the response, I am maintaining my score.

---

### Author Rebuttal · Authors · 2024-08-06

We thank AC and all reviewers for reviewing our submission and recognizing the contributions of our work. We are grateful for the valuable comments and suggestions.

### 1. Response to each review

For each review, we address the major question/concern in the rebuttal. We leave discussions and answers to minor questions in a comment appended.

### 2. Covariance matrix visualization

As suggested by Reviewer Mgdn, we provide the visualization results of the covariance matrices, collected from the three tasks MetaMath, NQ open, and Trivia QA, in **the PDF attached.** Please zoom in for a better view.

Since the original dimension in 4096 or 11008 will be too large to be informative, we downsample the covariance matrices into 32 $\times$ 32 and visualize their heatmaps. We provide the results from the activations before all the weights including ``self_attn.k_proj`` (the same as ``q_proj`` and ``v_proj`` due to the same input), ``self_attn.o_proj``, ``mlp.down_proj``, and ``mlp.gate_proj`` (the same as ``mlp.up_proj``) in the first layer, and the ``self_attn.o_proj`` weight in later layers.
We use red circles to mark the similar patterns, which the heatmaps from NQopen and TriviaQA share but do not appear in the one from the different task MetaMath.
The visualization result empirically supports that the covariance matrix patterns can be used to characterize the triggered task.

### 3. Common issue by Reviewer Mgdn and Reviewer 527n:

Reviewer Mgdn concerns about the impact of data amount and type to derive $C$ and how to choose $r$. Reviewer 527n has questions about scalability to other datasets and sample complexity.

- _Amount and type of data_:

**We respectfully disagree with the comment that "The amount and type/quality of data used to derive the covariance matrix $C$ significantly impact performance/usability of CorDA".
Actually, we have conducted an analysis in our paper about the samples used to collect covariance matrix.**

(1) Amount

In Table 6, we analyze the impact of sample number. We compare the results of collecting 32 and 256 samples for Wikitext-2 and PTB, respectively. In both Wikitext-2 and PTB, we observe a slight advantage of using 256 samples only when discarding the smallest 1024 ranks (6.35 v.s. 6.62 for Wikitext-2 and 22.28 v.s. 22.68 for PTB). When discarding the smallest 0-512 ranks, the results of using 32 and 256 samples are very similar and are much better than Plain SVD and ASVD. Therefore, only a few samples are enough to capture the task context by the covariance matrix. In order to also investigate the effect of sample number in the knowledge-preserved adaptation (KPA), we collect less samples (128 and 32) from NQopen and compare the resutls as follows.

|Samples|Trivia QA|NQ open|WebQS|GSM8k|Math|Avg|
|---|---|---|---|---|---|---|
| 256|44.30 |9.36 |7.14 |44.58 |6.92 |22.46|
| 128|44.53 |9.15 |7.16 |44.79 |6.85 |22.50|
| 32|44.11 |9.30 |6.94 |44.70 |6.93 |22.39|

Similarly, the sample number choice within [32, 256] does not cause a huge performance deviation. **Collecting covariance matrices from only a few samples are enough to implement our method.**

(2) Type/quality

The inputs/queries from a close task (e.g. questions from TriviaQA and NQopen) have a similar effect because they both trigger a similar ability. We have conducted analysis in our experiments. For the KPA mode, in Table 4 of our paper, the last two rows indicate that collecting data from TriviaQA and NQopen has a close performance on the QA benchmarks and the finetuning task. They are both much better than Plain SVD without considering any task context. For the IPA mode, as shown in Table 5, we collect covariance matrices from WizardLM-Evol-Instruct and Alpaca to build adapters, respectively, and finetune them on instruction following. They also lead to similar results (5.15 and 5.06) on MTBench, which are both better than the full finetuning, LoRA, and PiSSA results listed in Table 2. These results indicate that **randomly collecting context from one dataset has a scalable effect to other datasets of the same task.**

Moreover, in Table 4, we show the standard deviation of the results run with different seeds. It is shown that **randomly sampling data does not cause a large performance deviation**, which implies that it is not necessary to specially check the data quality.
Indeed introducing a data selection strategy when collecting the covariance matrix may bring further improvement, but we believe that this is an interesting extension of CorDA deserving our future exploration.

**Therefore, the effectiveness of our method is NOT sensitive to the data selection and does NOT rely on a large amount of data or specific type/quality.**

- _How to choose $r$:_

**Please note that $r$ is NOT a hyper-parameter introduced by our method.** It is just the low intrinsic dimension of the LoRA adapter. A higher $r$ has a better finetuning performance with more trainable parameters. A lower $r$ is more parameter-efficient but has lower performance.

The goal of our study is to introduce task context into the process of adapter initialization, and accordingly enable parameter-efficient finetuning with better world knowledge maintenance or stronger finetuning performance.
Therefore, **we follow the standard setting in LoRA, DoRA, and PiSSA**, _e.g._, usually setting $r=128$ for all linear layers.
It also facilitates fair comparison with these methods.  In our Figure 3 of our paper, we compare CorDA with full finetuning, LoRA, and PiSSA in both $r=128$ and $r=32$, to show the effectiveness of our method in different scenarios.

Therefore, we do not need to provide any guidance on how to choose $r$. Just the same as LoRA, DoRA, and PiSSA, $r$ is a configuration to control the tradeoff between efficiency and performance, determined by user's preference.

---

### Decision · Program_Chairs · 2024-09-25

**Decision:**

Accept (poster)

**Comment:**

The paper introduces a new PEFT method to better initialize LoRA parameters using a covariance matrix of input samples to preserve the context of the task. The authors claim that, by using input dependent covariance matrix, the model could be better adapted to either better preserving the world knowledge adaptation or achieving better instruction following adaptation. The authors compare the proposed method with various baselines and show that it is superior to the existing methods.

Summary Of Reasons To Publish:
The reviewers commonly appreciate that the paper is well-organized and clearly written. Also, the reviewers generally agree that the proposed method is novel. Reviewers agree that the experiments presented in the paper well supports the claim by comparing the baselines. There were common questions among the reviewers how much the method is sensitive to the amount and quality of the sample data. The authors addressed this issue by providing supporting experiments and the reviewers agreed.

Summary Of Suggested Revisions:
One request from reviewer kduk was to tone down the theoretical claim as it was not fully supported. The authors agreed on the point and promised they would tone it down.

All in all, this paper provides a new PEFT clearly.